# Motility, Biofilm Formation and Antimicrobial Efflux of Sessile and Planktonic Cells of *Achromobacter xylosoxidans*

**DOI:** 10.3390/pathogens8010014

**Published:** 2019-01-27

**Authors:** Signe M. Nielsen, Line N. Penstoft, Niels Nørskov-Lauritsen

**Affiliations:** 1Department of Clinical Medicine, Health, Aarhus University, DK-8200 Aarhus, Denmark; sinnel@rm.dk; 2Department of Clinical Microbiology, Aarhus University Hospital, DK-8200 Aarhus, Denmark; Line@thearmypainter.com

**Keywords:** *Achromobacter xylosoxidans*, cystic fibrosis, transcriptomics, biofilm, antimicrobial efflux

## Abstract

*Achromobacter xylosoxidans* is an innately multidrug-resistant bacterium capable of forming biofilms in the respiratory tract of cystic fibrosis (CF) patients. During the transition from the planktonic stage to biofilm growth, bacteria undergo a transcriptionally regulated differentiation. An isolate of *A. xylosoxidans* cultured from the sputum of a CF patient was separated into sessile and planktonic stages in vitro, and the transcriptomes were compared. The selected genes of interest were subsequently inactivated, and flagellar motility was found to be decisive for biofilm formation in vitro. The spectrum of a new resistance-nodulation-cell division (RND)-type multidrug efflux pump (AxyEF-OprN) was characterized by inactivation of the membrane fusion protein. AxyEF-OprN is capable of extruding some fluoroquinolones (levofloxacin and ciprofloxacin), tetracyclines (doxycycline and tigecycline) and carpabenems (ertapenem and imipenem), which are classes of antimicrobials that are widely used for treatment of CF pulmonary infections.

## 1. Introduction

*Achromobacter* species are emerging pathogens in cystic fibrosis (CF) [1,2,3]. Antimicrobial treatment of *Achromobacter* is challenging due to innate and acquired resistance towards a wide range of antimicrobial agents [4,5,6,7]; moreover, these bacteria have the capacity for biofilm formation [8,9,10], which is critical for antimicrobial tolerance. The transition from planktonic to sessile growth is an environmentally driven process induced by external factors [11]. The characteristics of bacterial cells embedded in biofilms include the lack of motility, excretion of extracellular matrix components, increased activity of efflux pumps and altered metabolic processes [11,12,13,14,15]. The achievable back-transition to the planktonic phase indicates that sessile growth is regulated by temporary, reversible alterations, rather than persistent mutations in the genome [11,16,17].

The gene expression profile of *Pseudomonas aeruginosa* during planktonic growth differed from the expression in in vitro biofilms subjected to experimental stress [17,18]. We have previously characterized the progression of gene regulation from early to late isolates cultured from a single patient chronically infected with *Achromobacter xylosoxidans* [19]. In the present study, we perform the transcriptomic analysis of a single isolate cultured during biofilm and planktonic growth phases in vitro. Noticeable differences are further characterized by the assessment of biofilm formation and the antimicrobial susceptibility of knockout mutants.

## 2. Results and Discussion

### 2.1. Differentially Expressed Genes

AX08 was propagated at sessile and early stationary planktonic growth phases in vitro. The strain is the “intermediate isolate” (CF-2b) that was used to characterize *Achromobacter* biofilm morphology [10] and gene expression [19]. Transcriptomic analysis during the separate conditions revealed a total of 232 differentially expressed genes, using a five-fold limit for significant difference. Annotation by RAST (Rapid Annotation using Subsystem Technology) [20] suggested that 89 genes encode hypothetical proteins, while 143 proteins of known or presumed functions were distributed into 16 subsystem categories; of these, 104 were upregulated and 39 were downregulated during sessile growth. The full list of 143 differentially regulated genes is presented in Appendix A; selected genes and regulatory adjustments are shown in Table 1.

#### 2.1.1. Increased Efflux Pump Activity During Sessile Growth

Efflux pumps are key mechanisms of resistance in Gram-negative bacteria, and many bacterial species harbor several RND-type multi-drug resistance pumps [21,22,23,24]. Four operons encoding RND (resistance-nodulation-division)-type multidrug efflux pumps designated MexAB-OprM, MexCD-OprJ, MexEF-OprN and MexXY-OprM have overlapping but different substrate specificities in *P. aeruginosa* [25]. Annotation of *Achromobacter* strain AX08 by RAST revealed the presence of similar operons in this species (not shown). Bador et al. demonstrated the involvement of AxyAB-OprM in *A. xylosoxidans* (with homology to MexAB-OprM) in resistance to cephalosporins (except cefepime), aztreonam, nalidixic acid, fluoroquinolones and chloramphenicol [26], whereas AxyXY-OprZ (with homology to MexXY-OprM) is responsible for the intrinsic high-level resistance of *A. xylosoxidans* to aminoglycosides [27]. When we compared the efflux pumps of successive isolates cultured at various time-points after the initial colonization, only AxyAB-OprM was significantly upregulated [19]. In *P. aeruginosa*, this efflux pump is characterized by the broadest-spectrum activity [25]. We compared gene regulation of a single isolate separated into planktonic or sessile growth phases, and observed a 7.4 times upregulation of *axyA* during sessile growth (Table 1). Thus, this efflux pump may be involved in *Achromobacter* biofilm metabolism, in addition to antimicrobial tolerance. Bacterial efflux pumps are not restricted to the extrusion of antimicrobial agents but can dispose solutes, metabolites, quorum sensing molecules and toxins [28,29,30]. The *axyB* gene of the AxyAB-OprM operon, also encoding the *axyA* gene found to be upregulated in AX08 biofilm, was successfully inactivated *Achromobacter* by Bador et al. [26]. Therefore, we decided to investigate another, yet uncharacterized, RND efflux pump of *Achromobacter* strain AX08 sharing closest similarity with MexEF-OprN of *P. aeruginosa*.

The efflux pump of *Achromobacter* is designated AxyEF-OprN, and *axyE* was selected for inactivation.

#### 2.1.2. Motility, Stress Response and Quorum Sensing

All 16 differentially expressed genes encoding flagellar motility were downregulated during biofilm growth (range: −8.5 to −5, Table 1). Stress response genes were generally upregulated in sessile AX08, but genes of the universal stress protein family (*uspA* and a tandem domain) were downregulated (Table 1 and Appendix A). For *E. coli*¸ inactivation of the universal stress proteins, UspC and UspE, abrogate motility [31]; the decreased expression of the *usp* genes in *A. xylosoxidans* during sessile growth suggests a similar involvement in this species. We observed an increased expression of the diguanylate cyclase/phosphodiesterase gene in sessile AX08 (Table 1). Cyclic-di-GMP acts as a quorum sensing secondary messenger in *Pseudomonas aeruginosa*, and elevated levels of cyclic-di-GMP, mediated by diguanylate cyclases and phosphodiesterases, are associated with decreased motility and formation of biofilm in that species [32,33]. The mechanisms of *Achromobacter* quorum sensing has not been examined, but our data suggest involvement of cyclic-di-GMP in the biofilm growth phase of *A. xylosoxidans*, possibly in quorum sensing signalling.

The flagellar M-ring protein, FliF (Table 1), is required for the synthesis of a functional flagellum [34], and *fliF* was selected for inactivation.

#### 2.1.3. Sulfur Metabolism

The most pronounced upregulation during sessile growth (86 fold) occurred in the cystathionine beta-lyase (*metC*) gene (Table 1). Anaerobic conditions occur in sputum from chronically infected CF patients, and *Achromobacter* can grow anaerobically by denitrification [35,36]. The high upregulation could indicate a role for sulfur metabolism during anaerobic growth of *Achromobacter* in vivo, with organic or non-organic sulfur components acting as electron acceptors. Furthermore, sulfur is required for the bacterial production of methionine. Sulfur is assimilated by the sulfate pathway, where sulfite oxidase oxidizes sulfite to sulfate, which is incorporated into cysteine. Cysteine functions as a sulfur donor in methionine biosynthesis, catalyzed by cystathionine beta-lyase (MetC) [37], and the *metC* gene has been shown to be of importance to the virulence of *Salmonella enterica* serovar Typhimurium [38,39]. 

Although anaerobic conditions are unlikely to prevail in a three-day-old biofilm, an increased expression of *metC* could potentially be a response to the decreased levels of oxygen and nutrients in the inner parts of the in vitro-grown biofilm. In our previous analysis of sequential isolates of the present strain isolated during a time-span of seven years [19], a putative sulfite oxidase gene was progressively upregulated, suggesting a role of the sulfate pathway for adaptation to long-term colonization.

*metC* was selected for inactivation.

#### 2.1.4. Constituents of the Extracellular Matrix

Exopolysaccharides are important constituents of the extracellular matrix [14,40]. Gene expression in biofilm showed an increase in genes affecting the formation and maintenance of the cell wall and capsule, and contributing to the formation of the extracellular matrix. Expression of the *epsF* gene, encoding exopolysaccharide biosynthesis, increased 6.4 times during sessile growth (Table 1), and the capsular polysaccharide ABC transporters, KpsT, KpsE and KpsM, increased 5.1 to 8.7 times during sessile growth (Table 1). The ABC transporter complex (KpsT/KpsM) exports polysaccharides across the cytoplasmic membrane [41] and, thereby, contributes to the increased generation of extracellular matrix components of *Achromobacter* biofilm.

Overproduction of alginate induces the mucoid phenotype and is a hallmark of chronic infection with *P. aeruginosa*, [42]. The alginate biosynthesis protein, AlgJ, is present in the genome of AX08. However, no increase in expression of this gene occurred during sessile growth, and alginate may not be an essential part in the generation of *Achromobacter* biofilm.

### 2.2. Examination of Inactivated Mutants

The investigation of the relevance of motility, cellular discharge and methionine biosynthesis for biofilm formation and antimicrobial susceptibility took place by the construction of knockout mutants (Δ*fliF*, Δ*axyE* and Δ*metC*, respectively), and inactivation of the selected genes were confirmed by whole-genome sequencing (not shown).

#### 2.2.1. RND-Type Efflux Pump AxyEF-OprN Affects Antimicrobial Susceptibility and Biofilm Formation

A two-fold or larger decrease in the minimal inhibitory concentration (MIC) was observed for selected agents of three antimicrobial classes, namely, carbapenems, flouroquinolones and tetracyclines (Table 2); with respect to levofloxacin, the MIC was reduced below the interpretative criterion defined by EUCAST for *Pseudomonas* spp. (≤ 1 mg/L, [43]), rendering the isolate susceptible to this agent. Unexpected increases in MIC were observed for the β-lactam antimicrobial agents, ceftazidime (2-fold) and doripenem (4-fold), which belong to the cephalosporin- and carbapenem-class of β-lactams, respectively (Table 2); this opposite effect was reproduced in three independent experiments.

Thus, the AxyEF-OprN efflux pump may be involved in the innate antimicrobial resistance of *A. xylosoxidans*. The extrusion of antimicrobials was limited compared with the broader spectrum targeted by AxyXY-OprZ, and particularly AxyAB-OprM [20,21]. A narrow substrate specificity spectrum also characterizes the MexEF-OprN efflux pump of *P. aeruginosa*, which exhibits the narrowest spectrum of the RND efflux pumps of that species [30]. Further studies are needed to elucidate the biological role of AxyEF-OprN in *A. xylosoxidans*.

A clinical isolate of *A. xylosoxidans* resistant to meropenem with increased levels of a novel betalactamase with carpabanemase activity, Axc, and an amino acid substitution at position 29 of the *axyZ* gene (V29G) suggests that axc expression is regulated by *AxyZ* [44]. The putative TetR family transcriptional regulator, encoded by the *axyZ* gene, functions as a negative regulator of the AxyXY-OprZ efflux pump. The exposure to tobramycin in strains with this particular mutation led to the overproduction of the AxyXY-OprZ efflux pump, resulting in increased MIC towards a range of antimicrobials including aminoglycosides, fluoroquinolones and tetracyclines [45]. The mechanisms of acquired resistance in *Achromobacter* may have implications for treatment strategies, warranting further investigation of *Achromobacter* resistance mechanisms. 

The ΔAxyE mutant was characterized by a slight decrease in biofilm formation (16%) measured as adherence to, and subsequent growth on, abiotic surfaces (Figure 1). However, this insubstantial decrease is not considered clinically relevant and no visual difference in biofilm morphology was demonstrated (Figure 2).

The relatively undisturbed formation of biofilm is in accordance with previous results showing no effect of the general efflux pump inhibitor, phenylalanine arginyl β-naphthylamide, on *A. xylosoxidans* biofilm formation [19]. Since inactivation of *axyE* reduced the MIC of levofloxacin, rendering the mutant sensitive to this compound, the effect on antimicrobial susceptibility towards levofloxacin in biofilm was likewise tested. However, no reduction in the minimal biofilm eradication concentration (MBEC) was observed (not shown).

#### 2.2.2. Motility Impairment Affects Biofilm Formation

The non-motile character of sessile growth was reflected by the downregulation of all genes involved in flagellar motility (Table 1 and Appendix A). However, flagellar motility is required for the inception of the biofilm [33]. Inactivation of the flagellar M-ring protein, FliF, resulted in a 39% reduction of in vitro biofilm formation on peg-lids (Figure 1), and visualization by confocal laser scanning microscopy (CLSM) demonstrated an almost complete lack of biofilm formation (Figure 2). The importance of flagellar motility for biofilm formation has been demonstrated in other bacterial species including *P. aeruginosa* [33,46,47]. Whether flagellar motility affects the formation of *Achromobacter* aggregates in CF sputum, in vivo, remains to be studied.

#### 2.2.3. Methionine Biosynthesis

Methionine biosynthesis depends on the assimilation of sulfur by the sulfate pathway, where cysteine functions as a sulfur donor in methionine biosynthesis, catalyzed by cystathionine beta-lyase (MetC) [37]. Due to the hyperexpression of this gene during sessile growth, we speculated that MetC, besides playing a putative role in anaerobic respiration, might be important for the establishment of *Achromobacter* biofilm. Cysteine is a precursor of methionine, and inactivation of CymR, the master regulator of cysteine metabolism in *S. aureus*, plays a role in biofilm formation in this species [48]. However, inactivation of the *metC* gene (Δ*metC*) in AX08 did not affect the amount or morphology of biofilm on abiotic surfaces (Figure 1 and Figure 2), or the planktonic growth rate (Appendix A). Possibly, the nutrient-rich Brain Heart Infusion (BHI) growth media could have masked potential effects, and the putative role of MetC in *Achromobacter* biofilm formation is unresolved. Additional experiments using a minimal growth media and anoxic biofilm growth conditions are required to further clarify the role of MetC in *Achromobacter* biofilm.

In conclusion, we characterized an additional RND efflux pump in *A. xylosoxidans* designated AxyEF-OprN. Despite a relatively narrow spectrum of extrusion of antimicrobial agents, deletion of the linker protein of this pump rendered the isolate susceptible to levofloxacin. The disruption of flagellar motility caused an almost complete nullification of biofilm formation. Transcriptomic analysis highlighted several subsystems and genes of importance for sessile growth in *A. xylosoxidans*.

## 3. Materials and Methods

### 3.1. Clinical Strain of A. xylosoxidans

The strain designated AX08 corresponds to the “intermediate isolate” of *Achromobacter xylosoxidans* cultured 1 year after initial colonization [19]. This isolate was characterized by a pronounced adherence to abiotic surfaces (also designated isolate CF2-b in [10]). The isolate was routinely cultured on 5% blood agar at 37 °C.

### 3.2. Culture at Separate Growth Phases and RNA Extraction

Planktonic cultures were generated by overnight incubation with shaking at 180 rpm of a single colony transferred to 10 ml of BHI media. The early stationary growth phase was prepared by the dilution of overnight cultures to an OD_600_ of 0.1 (corresponding to approximately 10^6^ cells/mL) and the transfer of 1 mL to Erlenmeyer flasks containing 10 ml of BHI media in triplicate. The cultures were grown with shaking at 180 rpm to an OD_600nm_ of 0.8, which corresponds to the early stationary growth phase (Appendix A), and then stabilized by treatment with RNA Protect (RNeasy Protect Bacteria Mini Kit, Qiagen, Hilden, Germany) according to the manufacturer’s recommendations. The preparation of biofilms in six-well plates and the extraction of RNA were performed as described [19].

RNA was purified using the RNeasy Protect Bacteria Mini Kit (Qiagen) according to the manufacturer’s protocol for enzymatic lysis and the proteinase K digestion of bacteria, with the lysis time extended to 30 minutes. The samples were treated twice with RNAprotect for 15 minutes. Turbo DNase (Ambion) treatment was used to degrade DNA. The integrity of the RNA samples was evaluated by Qubit ™ Fluorometric Quantitation (Thermo-Fisher Scientific, Waltham, MA, USA) and treated with Ribo-Zero™ rRNA Removal Kit (Bacteria) (Illumina, San Diego, CA, USA) to remove rRNA prior to sequencing

### 3.3. Sequencing and Data Processing

The cDNA preparation, sequencing and data processing of mRNA extracted from separate growth phases were performed as previously described [19], using ScriptSeq™ Complete Kit (Bacteria)–Low Input (Illumina), the Illumina NextSeq 500 platform generating 150 bp long paired-end reads, and the CLC Genomics Workbench RNAseq tools (https://www.qiagenbioinformatics.com/).

### 3.4. Gene Inactivation

The pUC19 vector is incapable of replication in *Achromobacter* [26] and was used as a suicide vector. pUC19 encodes a carboxy-penicillinase, and strain AX08 is innately susceptible to ticarcillin. The primers for PCR were designed to amplify approximately 800 bp of the genes, and included 15 bases of homology with the ends of the linearized pUC19 vector (Table 3).

The plasmids were constructed using the In Fusion^®^ HD cloning kit (Clonetech Laboratories, Inc. USA) and transformed into competent *Escherichia coli* Stellar™ cells according to the manufacturer’s protocols. The transformants were selected on LB plates containing 50 µg/mL of ticarcillin. The plasmids were purified using a Plasmid Purification Midi kit (Qiagen). 

AX08 cells were cultured overnight in BHI to an OD_600nm_ of between 1 and 1.5. Forty milliliter cultures were transferred to a 50 mL Falcon tube and cooled on ice for 20 minutes. For the generation of electro-competence, the cells were pelleted by centrifugation at 3500 rpm for 10 minutes at 4 °C, resuspended in 40 mL of 0.3 M sucrose and centrifuged again for at 3500 rpm for 10 minutes at 4 °C. After being washed in sucrose buffer three times, the cells were resuspended in 400 µL of sucrose buffer and placed on ice for 1–2 hours.

The plasmids were introduced by electroporation. Sixty milliliters of electro-competent AX08 and five microliters of plasmid (1–5 µg DNA) were mixed in an Eppendorf tube and left on ice for ten minutes. The suspension was transferred to an ice cold 1-mm electroporation cuvette (Cuvettes Plus™ Electroporation Cuvettes, BTX™, VWR, Radnor, Pennsylvania, USA) and electroporated using a Bio Rad MicroPulser ™ (Bio Rad, Hercules, CA, USA) at 1.5 kV for approximately 5 ms. Immediately thereafter, one milliliter of pre-heated (37 °C) LB media was added to the cuvette and gently mixed with the bacterial suspension. The mixture was transferred to an Eppendorf tube and incubated for two hours at 37 °C with shaking at 120 rpm. The recombinant bacteria were selected by plating 200 µL on LB agar plates containing 50 µg/mL of ticarcillin. The remaining 800 µL of bacterial suspension was centrifuged at 6000 rpm for five minutes. The supernatant was removed and the pellet was resuspended in 200 µL of growth media and plated on LB agar plates containing 50 µg/mL of ticarcillin. The plates were incubated for two days at 37 °C. The insertion of the target gene into the pUC19 vector was confirmed by the amplifying target sequences of the insert and the vector, and the insertion of the plasmid into the target gene was verified using whole-genome sequencing of the mutants. The libraries for whole-genome sequencing were prepared from 1 ng gDNA using Nextera XT (Illumina) and were subsequently sequenced on a NextSeq 500 (2 × 150 bp, Illumina).

### 3.5. Visualization and Quantitation of Biofilm Formation

The overnight cultures in BHI broth were adjusted to an OD_600nm_ of 0.1, and 180 µL were transferred in triplicate to wells in a 96-well plate with a glass bottom constructed for confocal microscopy (Ibidi µ-Plate 96 well, Ibidi, GmbH, Martinsried, Germany). The plates were incubated for 72 hours at 37 °C; 100 µL of BHI was carefully replaced with fresh broth in each well after 24 and 48 h avoiding agitation of the biofilm developing at the bottom. The biofilms were stained with LIVE/DEAD® stain (BacLight L7007 bacterial viability kit for microscopy, Invitrogen, Thermo Fisher, Waltham, MA USA) according to the manufacturer’s protocol, except that an increased concentration of propidium iodide (0.05 mM) was used for the staining of extracellular DNA [49]. The biofilms were imaged by confocal laser scanning microscopy (CLSM) (Zeiss LSM 700, Carl Zeiss AG, Oberkochen, Germany). The biofilm formation was quantified using the crystal violet microtiter assay, modified and adapted for adherence to peg-lids [50,51], as previously described [10].

### 3.6. Antimicrobial Susceptibility

The minimal inhibitory concentration (MIC) was determined for 21 antimicrobial agents using Sensititre GNX2F plates incubated for 20 h at 37 °C and analyzed by the Sensititre® Windows Software SWIN® (Termo-Fischer Scientific, Waltham, MA, USA) according to the manufacturer’s recommendations. The minimal biofilm eradication concentration (MBEC) of levofloxacin was determined as described previously [10].

## Figures and Tables

**Figure 1 pathogens-08-00014-f001:**
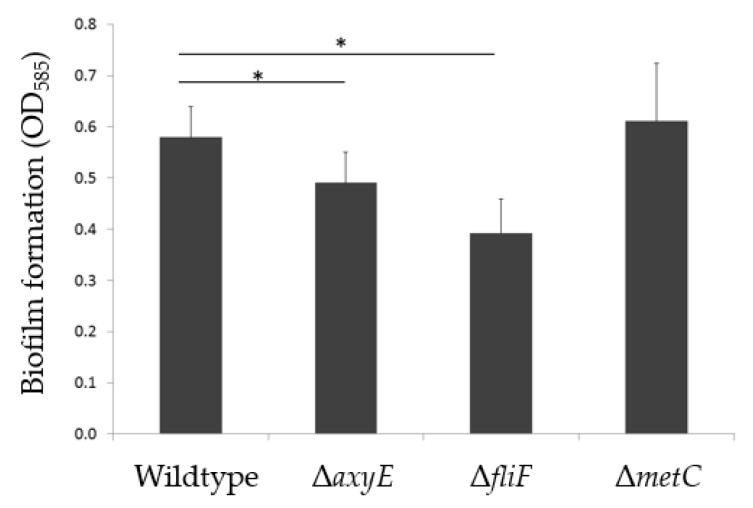
Mean production of biofilm in *A. xylosoxidans* AX08 and knockout mutants of the RND efflux system (Δ*axyE*), flagellar M-ring protein (Δ*fliF*) and methionine biosynthesis (Δ*metC*).

**Figure 2 pathogens-08-00014-f002:**
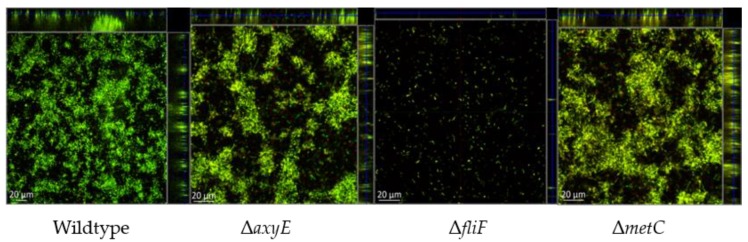
Visualization by confocal laser scanning microscopy (CLSM) of static biofilms in wildtype *A. xylosoxidans* AX08 and knockout mutants of the RND efflux system (Δ*axyE*), flagellar M-ring protein (Δ*fliF*) and methionine biosynthesis (Δ*metC*). Representative images from three biological replicates.

**Table 1 pathogens-08-00014-t001:** Selected regulatory adjustments during sessile growth.

Gene Name	Function	Fold Change	
RND efflux system membrane fusion protein *AxyA*	Virulence	Up	7.4
Flagellar basal-body rod modification protein *FlgD*	Flagellar motility	Down	−7.3
Flagellar basal-body rod protein *FlgC*	Flagellar motility	Down	−5.1
Flagellar biosynthesis protein *FlhB*	Flagellar motility	Down	−5.1
Flagellar biosynthesis protein *FliC*	Flagellar motility	Down	−8.5
Flagellar biosynthesis protein *FliL*	Flagellar motility	Down	−6.7
Flagellar biosynthesis protein *FliQ*	Flagellar motility	Down	−5.2
Flagellar biosynthesis protein *FliT*	Flagellar motility	Down	−5.2
Flagellar hook-associated protein *FlgK*	Flagellar motility	Down	−5.2
Flagellar hook-associated protein *FlgL*	Flagellar motility	Down	−5.3
Flagellar hook-associated protein *FliD*	Flagellar motility	Down	−6.0
Flagellar hook-basal body complex protein *FliE*	Flagellar motility	Down	−5.4
Flagellar L-ring protein *FlgH*	Flagellar motility	Down	−5.0
Flagellar motor switch protein *FliM*	Flagellar motility	Down	−5.7
Flagellar motor switch protein *FliN*	Flagellar motility	Down	−7.3
Flagellar M-ring protein *FliF* ^a^	Flagellar motility	Down	−6.3
Flagellar protein *FliJ*	Flagellar motility	Down	−5.9
Universal stress protein family (tandem domain)	Stress response	Down	−9.2, −6.1
Universal stress protein *UspA*	Stress response	Down	−6.6
Diguanylate cyclase/ phosphodiesterase	Stress response	Up	11.4
Cystathionine beta-lyase, *MetC* ^a^	Methionine biosynthesis	Up	86
Exopolysaccharide biosynthesis glycosyltransferase, *EpsF*	EPS biosynthesis	Up	6.4
ATP-binding proteins, K*psT, KpsE, KpsM*	Cell wall and capsule	Up	5.1 to 8.7

^a^ Selected for gene inactivation.

**Table 2 pathogens-08-00014-t002:** The minimal inhibitory concentrations (MICs) of AX08 and ΔaxyE.

	AX08 Wildtype		AX08 Δ*axyE*	
Antibiotic (µg/mL)	MIC	S/R *	MIC	S/R *
Colistin	1	S	1	S
Polymyxin B	1	NI	1	NI
Piperacillin	8	S	8	S
Ticarcillin/Clavulanic Acid	≤ 16	S	≤ 16	S
Cefepime	8	S	8	S
Cefotaxime	> 32	R	> 32	R
Ceftazidime	4	S	8	S
Aztreonam	> 16	R	> 16	R
Ertapenem	1	NI	0.5	NI
Doripenem	0.25	S	1	S
Imipenem	2	S	≤ 1	S
Meropenem	≤ 1	S	≤ 1	S
Ciprofloxacin	2	R	1	R
Levofloxacin	2	R	≤ 1	S
Amikacin	> 32	R	> 32	R
Gentamicin	> 8	R	> 8	R
Tobramycin	> 8	R	> 8	R
Doxycycline	8	NI	4	NI
Minocycline	≤ 2	NI	≤ 2	NI
Tigecycline	0.5	NI	≤ 0.25	NI
Trimethoprim/ Sulfamethoxazole	≤ 0.5	NI	≤ 0.5	NI

* Interpretation of susceptibility according to EUCAST clinical MIC breakpoints for Pseudomonas spp. S: Sensitive, R: Resistant, NI: No Interpretation.

**Table 3 pathogens-08-00014-t003:** Primers used for the generation of knockout mutants.

Primer	Nucleotide Sequence (5’ – 3’)	Fragment Length (bp)	Target Gene
FliF_F	CGGTACCCGGGGATCGAACAGATCAACTACCAGCG	771	Flagellar M-ring gene, *fliF*
FliF_R	CGACTCTAGAGGATCTTGATGTGGCTGATGGTG		
AxyE_F	CGGTACCCGGGGATCGACGTCAAGGAAAACCAG	795	RND-type multidrug resistance efflux pump gene, *axyE*
AxyE_R	CGACTCTAGAGGATCGGGCAACTGTTCGATCTT		
MetC_F	CGGTACCCGGGGATCTGATGCAGGACAAGGAGT	809	Cystathionine beta-lyase gene, *metC*
MetC_R	CGACTCTAGAGGATCTAGGCGTAGGTGTCGTAG		
M14F^a^	CCAGGGTTTTCCCAGTCACGA		
M14R^a^	GCGGATAACAATTTCACACAGGA		

Underlined sequences comprise 15 bases of homology with the ends of the linearized vector pUC19. ^a^ M14F and M14R primers are located in the vector adjacent to the insertion site.

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
