# Peer review of "Motility, Biofilm Formation and Antimicrobial Efflux of Sessile and Planktonic Cells of Achromobacter xylosoxidans"

_pathogens, 2019, doi:10.3390/pathogens8010014_

Round 1

Reviewer 1 Report

The work of Nielsen et. al. concerns the gene expression of Achromobacter xylisoxidans during two different lifestyles of growth, planctonic and sessile. The available respective data in the literature regarding this bacterium are limited, thus the present study is interesting and important.  This study complements the authors’ previous work on the differential regulation and expression of genes of Achromobacter xylosoxidans during biofilm formation during the time course of a CF patient chronic infection. The current study is well designed and the presented data are novel and of importance. The manuscript is well written and concise.

I have a few minor comments/suggestions:

1- The reader should go through the previous work to understand why the CF2-b “intermediate” isolate was chosen. My suggestion is to add a brief comment explaining in more detail the choice of this isolate.

2- Lines 74. Better to delete “stabilization of”.

3- Lines 151-158. It is apparent that further work is needed regarding the role of AxyEF-OprN efflux pump. The comparison with Salmonella enteric is not that relevant.

4- Lines 177-179. I consider the 16% decrease in biofilm formation as insubstantial.

5- MICs. The issue here is not the clinical breakpoints. To correlate differences in gene expression and antibiotic susceptibility levels it would be better to present the actual MIC values and not those offered by the limited range of Sensititre (optional). 

6- Tables 1 & 2 can be combined.

Author Response

Please see submitted Word file

Reviewer 2 Report

The manuscript of Nielsen et al. provide new information about the motility, biofilm formation ability and antimicrobial efflux of cells of A. xylosoxidans AX08 propagated at sessile and early stationary planktonic growth phases. This study was supported by several experimental techniques, such as transcriptomic analysis, inactivation of selected genes and study of the effect of the mutation in the ability to form biofilms and to resist to antimicrobials. It was observed that flagellar motility was essential for biofilm formation in vitro. It was also characterized a new RND efflux pump designated AxyEF-OprN. Transcriptomic analysis also revealed other genes of importance for sessile growth, such as metC gene and constituents of the extracellular matrix.

Major comments:

1)    Page 3, line 76 – 79. It was observed by transcriptomic analysis that the AxyA gene was upregulated in sessile cells. However, it was further study another efflux pump, the axyEF-OprN. The obtained results were interesting, leading to the characterization of a new RND efflux pump of A. xylosoxidans. However, the reason for this further characterization, not associated with the results of the transcriptome (the other 2 inactivated genes had significant changes in the transcriptome) is not well explained in the article. So please, rephrase this part of the manuscript.

2)    Page 6, line 219-221: Additional experiments using other growth media, such as minimal media, are required to confirm that the null effect of the metC inactivaction on biofilm formation and the planktonic growth rate was not masked by the nutrient-rich BHI growth media.

Minor comments:

1)    Please confirm the order of the references on the text, the references [20] to [25] were not correctly introduced on the text.

2)    Page 3, line 71 to 73: please rephrase the text.

3)    Page 3, line 73: please change “table S1” to “table 1”.

4)    Page 3, line 77: “, we chose to inactivate an additional operon …”. Why additional? The isolate chosen for inactivation had already a mutation in other efflux pump?

5)    Page 3, line 78: please change “if” to “of”.

6)     Page 3, line 85: please change “table 1” to “table 1 and S1”.

7)    Page 3, line 111: please change “deceased” to “decreased”.

8)    Table 2, footnote: remove “Table 2. MIC’s of AX08 and axyE”.

9)    Page 6, line 194-195: please rephrase the text.

10)  Page 11, reference 46: the date that was assessed the website is not correct, cannot be 2019.

Round 2

Reviewer 2 Report

The new version of the manuscript Nielsen et al. is more informative to reader, than the previous version. The authors addressed properly to my concerns.  

Minor correction:

Line 156 - Remove one of the "by".